# Molecular Typing Reveals Environmental Dispersion of Antibiotic-Resistant Enterococci under Anthropogenic Pressure

**DOI:** 10.3390/antibiotics11091213

**Published:** 2022-09-07

**Authors:** Anca Farkas, Cristian Coman, Edina Szekeres, Adela Teban-Man, Rahela Carpa, Anca Butiuc-Keul

**Affiliations:** 1Department of Molecular Biology and Biotechnology, Faculty of Biology and Geology, Babeș-Bolyai University, 1 M. Kogălniceanu Street, 400084 Cluj-Napoca, Romania; 2Centre for Systems Biology, Biodiversity and Bioresources, Babeș-Bolyai University, 5–7 Clinicilor Street, 400006 Cluj-Napoca, Romania; 3National Institute of Research and Development for Biological Sciences (NIRDBS), Institute of Biological Research, 48 Republicii Street, 400015 Cluj-Napoca, Romania; 4Department of Taxonomy and Ecology, Faculty of Biology and Geology, Babeș-Bolyai University, 1 M. Kogălniceanu Street, 400084 Cluj-Napoca, Romania

**Keywords:** antimicrobial resistance, ERIC-PCR, *Enterococcus avium*, *Enterococcus faecalis*, *Enterococcus faecium*, hospital, wastewater, freshwater

## Abstract

As a consequence of global demographic challenges, both the artificial and the natural environment are increasingly impacted by contaminants of emerging concern, such as bacterial pathogens and their antibiotic resistance genes (ARGs). The aim of this study was to determine the extent to which anthropogenic contamination contributes to the spread of antibiotic resistant enterococci in aquatic compartments and to explore genetic relationships among *Enterococcus* strains. Antimicrobial susceptibility testing (ampicillin, imipenem, norfloxacin, gentamycin, vancomycin, erythromycin, tetracycline, trimethoprim-sulfamethoxazole) of 574 isolates showed different rates of phenotypic resistance in bacteria from wastewaters (91.9–94.4%), hospital effluents (73.9%), surface waters (8.2–55.3%) and groundwater (35.1–59.1%). The level of multidrug resistance reached 44.6% in enterococci from hospital effluents. In all samples, except for hospital sewage, the predominant species were *E. faecium* and *E. faecalis*. In addition, *E. avium*, *E. durans*, *E. gallinarum*, *E. aquimarinus* and *E. casseliflavus* were identified. *Enterococcus faecium* strains carried the greatest variety of ARGs (*bla*_TEM-1_, *aac(6′)-Ie-aph(2″)*, *aac(6′)-Im*, *vanA*, *vanB, ermB*, *mefA*, *tetB*, *tetC*, *tetL*, *tetM*, *sul1*), while *E. avium* displayed the highest ARG frequency. Molecular typing using the ERIC2 primer revealed substantial genetic heterogeneity, but also clusters of enterococci from different aquatic compartments. Enterococcal migration under anthropogenic pressure leads to the dispersion of clinically relevant strains into the natural environment and water resources. In conclusion, ERIC-PCR fingerprinting in conjunction with ARG profiling is a useful tool for the molecular typing of clinical and environmental *Enterococcus* species. These results underline the need of safeguarding water quality as a strategy to limit the expansion and progression of the impending antibiotic-resistance crisis.

## 1. Introduction

Today, more than half of the world’s population lives in urban areas, a proportion expected to increase to 68% by 2050, according to the United Nations (https://www.un.org/, accessed on 31 July 2022). Not only are the cities themselves expected to be highly impacted by excessive anthropogenic pressure, but also the surrounding environments. The urban–rural lifestyle in metropolitan areas is a developing phenomenon, concerning the essential human activities and services, as well as recreation and leisure. As a consequence of global demographic challenges, both the artificial and the natural environment are increasingly affected by contaminants of emerging concern. The quality of water resources is vulnerable to a wide range of microbial pollutants, such as bacterial pathogens and their antibiotic-resistance genes (ARGs).

Aquatic environments are an ideal setting for the acquisition and dissemination of antimicrobial resistance, and human exposure to antibiotic-resistant bacteria and ARGs through water may pose an additional health risk [1]. *Enterococcus* species have frequently been described as carriers of antibiotic resistance across the One-Health continuum. Hospital effluents [2], untreated sewage [3] and raw manure [4] have been identified as the main hotspots for antibiotic-resistant enterococci and sources for their environmental spread. Following contamination events, enterococci can persist for long periods of time in different environmental matrices [5,6]. *Enterococcus* bacteria are all the more dangerous as potential vectors for antimicrobial resistance when escaping wastewater treatment. They pose a constant microbiological risk in surface waters that receive treated wastewaters [3,7] and continue to spread further downstream. Antibiotic-resistant enterococci have entered the groundwater environment, being isolated from untreated drinking water springs and wells [8,9], alluvial groundwater [10] and karst aquifers [11].

Enterococci are Gram-positive bacteria belonging to the phylum *Bacillota* (synonym Firmicutes), *Bacilli* class, *Lactobacillales* order, *Enterococcaceae* family. About 58 *Enterococcus* species have been recognized so far [12]. Molecular clock estimation, together with analysis of their ecology and phenotypic diversity placed the origins of the *Enterococcus* genus 500 million years ago, around the time of animal terrestrialization. Speciation occurred along with the diversification of hosts [13], enterococci being regarded as typically commensal bacteria for a long time. Essential members of animal microbiomes, they colonize mainly the digestive and urinary tracts. In humans, enterococci are found in concentrations of approximately 10^6^ to 10^7^ in the intestine (up to 1% of the colon microbiota) [14]. The most frequent *Enterococcus* species in human gastrointestinal tract are *E. faecalis* and *E. faecium*, followed by *E. casseliflavus* and *E. gallinarum* [15], along with *E. durans*, *E. hirae*, *E. avium* and *E. caccae*, which are less common [16].

From an evolutionary perspective, coevolution between bacteria and animals has selected intrinsic properties in enterococci, conferring them abilities to evade host defenses, compete in the intestinal tract, persist and spread in the environment. Remarkably resilient organisms, they are able to adapt to a broad range of pH, salinity and temperatures, survive sunlight exposure, desiccation, nutrient starvation, disinfection [13,17,18], microgravity and increased cosmic radiation [19]. Therefore, enterococci are able to disseminate into the environment and survive outside the animal body, being widely used as fecal indicators in water quality monitoring. The main sources of enterococci in natural environments include sewage, agricultural and urban runoff, animal manure, wildlife waste and bather shedding. During water quality monitoring, intestinal enterococci have been found in biofilms, even in drinking water systems providing safe tap water [20]. *Enterococcus* species are able to persist in stable microcolonies for long periods of time, entering a viable nonculturable state [21]. Even with the availability of modern molecular techniques, it is still difficult to decide what populations are part of the natural or transient microbiota of the environment.

Enterococci began to emerge as leading causes of multidrug-resistant hospital-acquired infections. When pathologic changes result through direct toxin activity, or indirectly triggering inflammatory damages, certain *Enterococcus* species may become responsible for human infections [22]. According to recent data, the *Enterococcus* genus is responsible for 10.9% of nosocomial infections in the EU/EEA region [23]. The most important pathogens are *E. faecalis* and *E. faecium*, but non-faecium non-faecalis enterococci, such as *E. avium*, *E. caccae*, *E. casseliflavus*, *E. dispar*, *E. durans*, *E. gallinarum*, *E. hirae* and *E. raffinosus*, have been increasingly reported to cause human infections [24]. *Enterococcus faecium* and *E. faecalis* have evolved to become globally disseminated nosocomial pathogens. Hospital-associated *E. faecium* strains are characterized by the acquisition of adaptive genetic elements, including genes involved in antibiotic resistance. In contrast to *E. faecium*, clinical variants of *E. faecalis* are not exclusively found in hospitals but are also present in healthy individuals and animals [25]. The apparent adaptations found in hospital-associated *E. faecalis* lineages likely predate the “modern hospital” era, suggesting selection in a different niche and underscoring the generalist nature of this nosocomial pathogen [26]. Very few Romanian studies concerning antimicrobial resistance of enterococci have been published. Clinical variants of *Enterococcus* showed a high resistance profile for fluoroquinolones and penicillins [27,28], while bacterial clones from fishery lakes were highly resistant to macrolides [29]. Vancomycin resistance recently emerged in this One-Health continuum.

The aim of this study was to determine the extent to which anthropogenic contamination may contribute to the spread of antibiotic resistant enterococci in aquatic compartments and to explore genetic relationships within and between *Enterococcus* species. For this purpose, environments from low to high presumptive fecal contamination related to anthropic pressure were assessed to quantify the burden of intestinal enterococci and the levels of phenotypic and genotypic antimicrobial resistance in a collection of isolates. It was of particular interest to identify the strains and to characterize their genetic diversity under the hypothesis that similarity of DNA banding patterns may be linked to their antibiotic resistance and also to the type of water source. For this objective, the effectiveness of ERIC-PCR fingerprinting was evaluated for *Enterococcus* species and strain differentiation.

## 2. Results

### 2.1. Water Contamination by Enterococci

Water contamination by intestinal enterococci was investigated in different aquatic compartments with different degrees of anthropogenic pollution: groundwater (GW1-GW4), surface waters (SW1-SW3), wastewater influents (WWI) and effluents (WWE), and hospital effluents (HE). Enterococci were detected in all samples, except for a groundwater well, in a range from 3 ± 2 colony forming units (CFU)/100 mL in a groundwater spring located outside the city area (GW1) to (465 ± 0.2) × 10^3^ CFU/100 mL in WWI. Sewage treatment contributed significantly to the reduction of microbial counts, to 99 ± 5 enterococci/100 mL in WWE. Hospital effluents harbored high concentrations of enterococci, but still below the loadings from municipal sewage. In surface waters, enterococci abundances increased along the river, from 9 ± 1 CFU/mL in SW1 to (11.7 ± 0.1) × 10^3^ in SW3. Groundwater samples were differently impacted by enterococcal contamination, which was found to be up to 80 ± 6 CFU/mL in GW2, a dug well from a village upstream of the city (Table 1). A dug well from Cluj city (GW4) was sampled three times, but since no intestinal enterococci were detected, it was excluded from further investigations.

### 2.2. Resistance to Antibiotics in Enterococci

Kirby–Bauer tests were performed for 547 *Enterococcus* isolates to identify their resistance to ampicillin (AMP), imipenem (IMP), norfloxacin (NOR), gentamicin (CN), vancomycin (VAN), erythromycin (E), tetracycline (TE) and trimethoprim-sulfamethoxazole (SXT). The overall prevalence of susceptible profiles was 41.5%. In all the sampling points where intestinal enterococci were detected, there were isolates displaying phenotypic resistance, and their proportions were between 8.2% and 94.4%. Variants of enterococci resistant to all the antimicrobial agents tested in this study were isolated from HE, WWE, WWI and SW3. Resistance up to eight antibiotics per strain was observed in hospital sewage, up to seven in wastewaters, up to three in river water (SW2 and SW3) and also in shallow groundwater wells (GW2 and GW3). Isolates from spring water (GW1) were resistant to a maximum of two antimicrobial drugs, while intestinal enterococci from drinking water source (SW1) to a single antibiotic.

The antibiograms indicated that 91.3%, 91.1% and 89.7% enterococci were susceptible to gentamycin, vancomycin and ampicillin, respectively. A total of 88.2% was susceptible to imipenem, 86.6% to norfloxacin and 84.8% to erythromycin. Tetracycline and trimethoprim-sulfamethoxazole were the least effective antimicrobial agents, inhibiting only 70.2% and 55.9% of *Enterococcus* isolates, respectively.

The magnitude of phenotypic resistance of intestinal enterococci categorized based on their origin is shown in Figure 1a. The highest frequency of antimicrobial resistance was observed against trimethoprim-sulfamethoxazole, in enterococcal isolates from WWI (92%), WWE (91%) and HE (50%). Tetracycline resistance was also high in all aquatic compartments, with many isolates from HE (51%), GW2 (50%) and SW2 (34%), WWI (38%) and WWE (43%) being resistant. Resistance against erythromycin was observed in all compartments, from 42% in HE to 1% in SW1. At a maximum frequency of 27% in HE, strains resistant to ampicillin were detected in all samples, except for SW1. Isolates resistant to norfloxacin, imipenem, gentamycin and vancomycin were present most frequently in hospital sewage (38%, 32%, 32%, and 30%) and in wastewater samples, but not always in surface waters and groundwater. All *Enterococcus* isolates from GW1 and GW2 were susceptible to these four antibiotics. Besides the SXT, TE, E resistance profiles and their combinations, the following most prevalent resistance patterns were NOR-SXT and AMP-IPM-NOR-CN-VAN-E-TE-SXT. From the 82 antibiotic-resistance patterns observed in 336 *Enterococcus* isolates, 48 patterns have only appeared once. Proportions of multidrug-resistant (MDR) enterococci were 44.6% in HE, 36.4% in WWE, 33.7% in WWI, 5.3% in SW2, 4.5% in GW2 and 2% in SW3. No MDR strains were present in SW1 and GW1. The overall frequency of MDR enterococci was 7.7%.

Detection by PCR of ARGs indicated that overall, 23.9% of enterococci investigated in this study (574 isolates) carried at least one of the targeted ARGs. The proportion of isolates with genetic-encoded resistance relative to the isolates displaying phenotypic resistance (336 strains) was 40.8%. The magnitude of genotypic resistance of intestinal enterococci categorized based on their origin is shown in Figure 1b. The greatest ARG diversity was observed in *Enterococcus* spp. from HE, where 9 out of the 17 target genes were detected. None of the investigated ARGs were detected in isolates from SW1, GW2 and GW3 sites. ARG relative frequencies were 0.58 in HE, 0.21 in WWE, 0.16 in WWI, 0.03 in SW3, 0.02 in SW2 and 0.01 in GW1. The most frequently detected were *tetM*, in 0.14% of *Enterococcus* isolates, followed by *tetL* (0.13%) and *ermB* (0.1%). The genes *tetL*, *tetM* and *ermB* were present in enterococci from 6 (SW2, SW3, GW2, WWI, WWE, HI), 4 (SW3, WWI, WWE, HE) and 3 (SW2, WWE, HE) out of 9 sampling sites, respectively. In addition, *bla*_TEM-1_, *aac(6′)-Ie-aph(2”)-Ia*, *vanA*, *vanB*, *tetB* and *tetC* were exclusively detected in HE. *Enterococcus* strains carrying the ARGs *aac(6′)-Im*, *mefA* and *sul1* as well as class 1 integron integrase *intI1* were only identified in wastewaters. The most prevalent ARG patterns were *tetL, ermB-tetL-tetM, tetM, tetL-tetM* and *sul1*. From the 36 ARG patterns observed, 19 had single appearances. PCR amplifications for *bla*_NDM-1_, *ermA*, *tetA*, *sul2* and *sul*3 had negative results. A moderate statistical significant correlation (R = 0.66) was found between the levels of displayed phenotypic resistance and the incidence of the investigated ARGs, suggesting that other genetic-encoded mechanisms might also be involved (Figure 2a).

### 2.3. Enterococcus Diversity and Association with Antibiotic Resistance Profiles

The molecular identification of 146 resistant isolates of *Enterococcus* has led to the recovery of seven taxons in different proportions: *E. faecium* (44.5%), *E. faecalis* (33.6%), *E. avium* (17.8%), *E. durans* (1.4%), *E. gallinarum* (1.4%), *E. aquimarinus* (0.7%) and *E. casseliflavus* (0.7%). Considering the water compartments where multiple species have been recovered from, resistant *E. faecalis* was found to predominate in WWI and SW3, *E. facium* in WWE and *E avium* in HE, respectively. From SW2, GW1 and GW2, all the resistant isolated were identified as *E. faecalis*. From SW1 and GW3, all the resistant isolates were *E. faecium*. Antibiotic-resistant *E. avium* was exclusively recovered from HE, and *E. aquimarinus* and *E. casseliflavus* from WWI, respectively (Table 1).

The genomic diversity analysis of 146 *Enterococcus* isolates was carried out using the repetitive sequence-based polymerase chain reaction (Rep-PCR) with the Enterobacterial Repetitive Intergenic Consensus (ERIC) primer ERIC2. Complex fingerprint patterns were found, consisting of 3 to 12 amplification bands. The genetic variation among isolates revealed different banding patterns, which ranged from 100 bp to 2 kb and 42% similarity. By applying the unweighted pair-group arithmetic mean method (UPGMA), dendrograms generated using Dice’s similarity coefficients were comparable and useful to study the intra- and inter-species diversity of *Enterococcus* isolates.

ERIC-PCR grouped all 26 *E. avium* isolates in two clusters and resolved 12 discrete genomic patterns. A similarity of 72% was found among *E. avium* isolates. Hospital effluent had a low *E. avium* diversity, most of the strains being closely related. Within cluster A, the REP-PCR profiles of 17 isolates were highly similar. In two subgroups, three isolates (HE-2, HE-14 and HE-46) and five isolates (HE-29, HE-38, HE-53, HE-54 and HE-68), respectively, had identical ERIC-PCR profile and also shared the same antibiotic resistance pattern, suggesting clonal relatedness (Figure 3).

Due to the high genetic similarity (85%), a typical ERIC-PCR fingerprint generated 16 distinct genomic patterns and grouped *E. faecalis* strains isolated from different water compartments. Among the 49 collected isolates, 45 isolates were grouped in 9 clusters. As expected, some isolates sharing the same origin clustered together, as observed for wastewater isolates in clusters D, E, F and G. However, segregation of the strains with respect to water matrices was not a general rule. The highly similar genetic patterns grouped *E. faecalis* isolates from hospital sewage, wastewater influents and effluents, from river water and groundwater together, in clusters A, B, C, H and I, despite their variability in the ARG profiles. Clonal relatedness was suggested by identical band pattern and also the ARG profile, as observed in two *E. faecalis* isolates from WWE within cluster B (WWI-12 and WWI-14), two isolates within cluster E (WWI-20 and WWI-65) and two isolates within cluster G (WWE-93 and WWE-100). In addition, within grouping I, the same ERIC-PCR and ARG profiles were observed for isolates collected from different matrices: WWI-59 and SW3-81 and also GW2-12, GW2-22, GW2-31, SW3-7, SW3-35 and SW3-53 (Figure 4).

Comparative analysis of Rep-PCR fingerprinting for the three main *Enterococcus* species revealed the most substantial genetic diversity among *E. faecium* strains. ERIC-PCR typing of 65 isolates resolved 35 discrete genomic patterns. However, bacterial isolates from different environmental compartments shared 68% genome similarity and hierarchical clustering grouped *E. faecium* strains in four main clusters. Similar to the *E. faecalis* typing results, *E. faecium* isolates sharing the same origin clustered together, but none of the ERIC-PCR patterns was exclusively specific for one aquatic regimen. The great variability of genetic patterns in grouping A was generated by strains from all aquatic compartments, with high diversity of antibiotic-resistance profiles. Clusters B and C mostly contained wastewater isolates, but also strains from groundwater. These strains displayed lower levels of antibiotic resistance, all the *E. faecium* from GW3 lacking the targeted genetic elements. Hospital effluents had a low *E. faecium* genetic diversity, most of the strains being clustered together in grouping D, together with a strain isolated from river water. Clonal relatedness suggested by identical ERIC-PCR and ARG profiles of *E. faecalis* isolates was observed within clusters A (SW1-22, SW1-35 and SW1-71), B (GW3-7, GW3-17 and GW3-37; GW3-32, GW3-33 and GW3-34; WW-1 and WWI-9) C (WWE-23 and WWE-26; WWE-12 and WWE-14; WWE-62 and WWE-63) and D (HE17 and HE-55; HE19, HE20 and HE50) (Figure 5).

Six isolates belonging to four other species (non-predominant species) were detected and characterized during this study. They shared 33% genetic similarity and generated six ERIC-PCR patterns (Figure 6). Clonal relatedness according to genetic typing was observed for *E. durans*, but the two variants had different resistance profiles. Two *E. gallinarum* isolates differed in both their ERIC-PCR band patterns and antibiotic-resistance profiles.

No statistically significant correlations were found between the number of banding patterns and the level of phenotypic or genotypic resistance (Figure 2b,c). Additional visualization tools were applied to infer the associations and differences between species. At the genus level, molecular typing revealed the clustering of *Enterococcus* isolates, both by ERIC-PCR profiles and by ARG patterns (Figure 7). Rep-PCR fingerprinting using the ERIC2 primer provided excellent discriminatory power at the species level within the genus *Enterococcus*, obvious in the UPGMA dendrogram. *Enterococcus faecium*, *E. avium* and *E. faecalis* strains clustered according to their taxonomy. Strains belonging to other species (*E. aquimarinus*, *E. durans*, *E. casseliflavus* and *E. gallinarum*) generated distinct band patterns, allowing their distinct differentiation in the UPGMA dendrogram (Figure 7a).

Some degree of clustering according to taxonomy was observed when the principal component analysis (PCA) was employed to provide an integrative view upon the ARGs involved in variation. Genetic variation of ARGs explained by the first two principal components (52.43% of the total variation) among *Enterococcus* species revealed slightly distinct groups (Figure 7b). Besides the isolates belonging to the group of non-predominant species, *E faecalis* strains harbored the lowest diversity of ARGs, and all their encoded-genetic traits were common to *E. avium* and/or *E. faecium* and also shared with other species. *Enterococcus avium* and *E. faecium* strains benefit as well from the ARG pool specific for the family, but differences in gene frequencies resulted in their clustering.

## 3. Discussion

*Enterococcus* species are valuable fecal indicators and important predictors of anthropogenic pollution and associated risks in aquatic environments. Waters with high enterococcal loads represent an environmental and a public health hazard, since most of these bacteria carry numerous antibiotic-resistance traits. However, molecular typing clustered the strains regardless of their source or antibiotic-resistance profile.

During this study, the microbiological risk associated with contaminated waters was correlated with the magnitude of exposure to anthropogenic pressure in various aquatic matrices. The detection and enumeration of intestinal enterococci in different water compartments confirmed that they are reliable indicators of water quality and that environmental reservoirs are closely related to human activities. The highest loads of intestinal enterococci were found in raw wastewaters and in hospital sewage. The performance efficiency of the municipal treatment plant, accounting for intestinal enterococci, demonstrated an average log reduction of 1.4. Unfortunately, high enterococcal contamination was found in the river basin and in several groundwater samples. Downstream of the wastewater treatment plant, the fecal pollution of river water was reflected in a 2-log increase in *Enterococcus* counts, compared to treated effluents. This suggests that besides discharge from the wastewater treatment plant, other anthropogenic activities have substantially contributed to river pollution. Across the city, accidental sewage spills, droppings from pets, littering, illicit dumping and waste disposal have been identified as the main sources of fecal microorganisms and nutrients in urban runoff, leading to the deterioration of the Someșul Mic River and its tributaries. Moreover, unanticipated high enterococcal loads were found in surface water and groundwater upstream of the city, where fecal contamination was mainly due to surface runoff from diffuse pollution sources. In these human impacted areas, the uncontrolled discharge of wastewaters and accidental sewage spills, animal farming practices, droppings from pets and wildlife, littering, illegal dumping and waste disposal, logging and sawmilling have been identified as the main sources of fecal microorganisms and nutrients in surface runoff, leading to the deterioration of the Someșul Rece River and Tarnița Reservoir. The water quality in shallow wells was largely affected by fecal contamination due to the infiltration of surface runoff, both in the rural (GW2) and urban areas (GW3 and GW4) but was also influenced individually by specific conditions (i.e., cleaning and disinfection practices). Proper construction and routine maintenance of dug wells are important to safeguard water quality, as observed for GW4, where intestinal enterococci were not detected during this study. The outcomes of the present project regarding contamination of the river continuum are consistent with previous findings. Antibiotics, antibiotic-resistant bacteria and their ARGs have been identified as contaminants of emerging concern in hospital effluents, wastewaters, surface waters [30,31,32,33] and groundwaters [34]. International guidelines and regulations enforce water quality surveillance based on monitoring of intestinal enterococci in conjunction with *E. coli*. Despite a decades-long attempt, no other more accurate and more reliable indicators have been found. Contaminated environments may serve as reservoirs of extra-intestinal enterococci. There is no consistent evidence of enterococcal regrowth within environmental biofilms, but apparently some *Enterococcus* species or strains are able to grow in extra-enteric compartments, developing potentially naturalized environmental populations. Vegetation was recently proved to promote bacterial regrowth in a warm climate, as submerged vegetation [35] and phytoplankton [36] for *E. casseliflavus*, eelgrass for *E. casseliflavus*, *E. hirae* and *E. faecalis* [37] or dune vegetation for *E. moraviensis* [38]. Modern molecular techniques may try to distinguish natural enterococcal populations from transient microbiota in the environment. Antibiotic resistant *E. faecium* and *E. faecalis* were largely found in aquatic habitats as the dominant species. *Enterococcus faecalis* was predominant in two surface waters (SW2 and SW3), two groundwater sources (GW1 and GW2), and in municipal wastewater. *Enterococcus faecium* was predominant in the Tarnița Reservoir, the main drinking water supply for Cluj County (SW1), a groundwater well within the city (GW3) and treated wastewater. An exception of particular significance is the predominance of *E. avium* in hospital effluents. A recent study investigating non-faecium and non-faecalis hospital infections in Cluj reported that *E. avium* seems to be involved more often in infectious neurological disorders, being the only species isolated from low respiratory tract infections [39].

The *Enterococcus* spp. are intrinsically resistant to a number of antimicrobials, including cephalosporins and sulfonamides, while they are only mildly resistant to β-lactams and aminoglycosides. Clinical strains with resistance to macrolides, tetracyclines, streptogramins and glycopeptides were described previously [40]. The proportion of MDR was higher in *E. faecium* (52%) compared to *E. faecalis* (51%), but not significantly different as expected. *Enterococcus avium* isolates also displayed a high level of multidrug resistance (43%). The variant of *E. gallinarum* isolated from hospital effluent was also MDR. Strains resistant to antibiotics for human use (VAN, CN, NOR, and IPM) were found mostly in hospital effluents, while resistance to antibiotics for veterinary use (TE, and E) was also observed in surface water and groundwater, even in less-impacted environments. Municipal wastewaters harbored enterococci with resistance to all classes of antimicrobial agents, reflecting the antibiotic consumption practices in the metropolitan area. Resistance to sulphonamides was exceptionally high in urban sewage, which is a common fact for wastewater treatment plants. Differences in the antibiotic-resistance profiles are known to reflect antimicrobial use practices in each country, region, or sector of the One-Health continuum (clinical, agricultural and environmental) [40].

Due to their ability to acquire antibiotic resistance determinants, multidrug-resistant enterococci display a wide repertoire of resistance mechanisms: the modification of drug targets, inactivation of therapeutic agents and overexpression of efflux pumps. The highest level of antibiotic resistance and the greatest diversity of ARGs was found in *E. faecium*, followed by *E. avium*, *E. faecalis* and *E. durans*. *Enterococcus faecium* isolates frequently carried *tetL*, *tetM*, *aac(6′)-Ie-aph(2”)-Ia*, *vanB*, *ermB* and *vanA* genes, and less frequently *sul1*, *tetC*, *tetB*, *mefA* and *bla*_TEM-1_. The ARGs detected in *E. faecalis* isolates were *tetM*, *tetL*, *ermB*, *sul1*, *aac(6′)-Ie-aph(2”)-Ia*, *vanA*, *vanB*, *tetB* and *tetC*. *Enterococcus avium* strains harbored the *ermB*, *tetM*, *tetL*, *aac(6′)-Ie-aph(2”)-Ia*, *vanA*, *tetB*, *tetC*, *vanB* and *bla*_TEM-1_ genes. The genes *ermB, tetL, tetM* and *sul1* were detected in *E. durans* isolates. The lowest diversity and prevalence of ARGs was found in *E. aquimarinus* (*tetL*), *E. casseliflavus* (*sul1*) and *E. gallinarum* (*tetL*) isolates. The acquisition of genes encoding vancomycin resistance is recognized as one of the features reflecting enterococci adaptability [41]. During this study, *E. avium*, *E. faecium* and *E. faecalis* harbored both the *vanA* and *vanB* genes. The gene *vanA* was more often present in *E. avium* and *E. faecalis*, while *vanB* in *E. faecium*. *Enterococcus* species are a serious health issue worldwide, particularly due to increasing vancomycin resistance and multidrug resistance (https://resistancemap.cddep.org/AntibioticResistance.php, accessed on 31 July 2022). In the European Region, during the past 10 years, vancomycin-resistant enterococci accounted for 1.1% of all pathogens isolated from patients with hospital-acquired infections. Among patients with hospital-acquired bloodstream infections with *Enterococcus* spp., mortality attributable to vancomycin resistant variants was 33.5% [23]. Last data from European Centre for Disease Prevention and Control reported vancomycin resistance to 3.3% in *E. facecalis* and 39.3% in *E. faecium* (http://atlas.ecdc.europa.eu/public/index.aspx, accessed on 31 July 2022). Our results are consistent with official reports, the gene *vanB* being detected 10 times more frequently in *E. faecium* than in *E. faecalis* isolates. At least the *vanA*, *vanB* and *aac(6′)-Ie-aph(2”)-Ia* genes, as markers of clinical enterococci, were exclusively detected in isolates from hospital effluents. The gene *bla*_TEM-1_ was detected in two multiresistant strains (*E. avium* and *E. faecium*), both from hospital sewage. This genetic feature needs further investigations, as beta-lactamases imply resistance mechanisms that are specific for Gram-negative bacteria. However, beta-lactamases were recently reported in Gram-positive bacteria [42], including the detection of *bla*_TEM-1_ in *E. faecalis* [43]. The results of this screening reveal that enterococci are important vehicles for both plasmid-borne and chromosomally encoded resistance determinants. They likely function as a reservoir of drug-resistance traits and can serve as vectors for the spread of these genes to other Gram-positive pathogens [41]. The horizontal gene transfer of mobile genetic elements is the major contributor to the emergence and dissemination of multidrug resistance. Class 1 integron integrase is a molecular marker for the mobilizable chromosomal ARG platforms and for anthropogenic pollution. The *intI1* gene was detected in two *E. faecium* and one *E. faecalis* strains, all isolated from the wastewater treatment plant. The *sul1* gene was also found exclusively in wastewater isolates, but no pattern of association was found between *intI1* and *sul1*. The *sul1* gene was detected in only one out of three strains carrying *intI1*. The linkage of the *intI1* integrase and *sul1* gene is a particularity of class 1 integrons in environmental Gram-negative bacteria [44].

DNA fingerprinting by ERIC-PCR is widely applicable since ERIC primers do not exclusively target enterobacterial repetitive elements [45]. It was demonstrated that it is a reliable tool, with high discriminatory power among *Enterococcus* strains isolated from food [46,47,48], water samples [49], clinical specimens obtained from animals [50,51] and humans [52]. In previous studies, the genotyping assay directed ERIC1 or ERIC1 in combination with ERIC2 primers against *E. faecalis* and/or *E. faecium* genomes. For the first time, the present study provides an optimized method, using only the ERIC2 primer, which allows discrimination among seven *Enterococcus* species and offers a better overview of their diversity. The genetic typing of *Enterococcus* isolates during this study generated significant results, in agreement with previous knowledge. One particular situation worth special attention, regarding the clustering of an *E. faecium* strain from surface water (SW3-39) in the same clade with the hospital-derived variants. The segregation between commensal enterococci and hospital-adapted lineages has been partly elucidated, and it is clearer for *E. faecium* than for *E. faecalis* [25]. It is known that *E. faecium* has a defined clade that diverged about 75 years ago and is associated with human infections, being rarely encountered in healthy individuals and even less in the environment. These clinical clones are characterized by hypermutability, increase in mobile genetic elements and alterations in metabolism [53]. In contrast to *E. faecium*, clones of *E. faecalis* isolated from clinical specimens are not exclusively found in hospitals, being also present in healthy individuals and animals. Molecular epidemiology using ERIC-PCR fingerprinting showed that *E. faecium* and *E. faecalis* isolates from different aquatic matrices exhibit the same or similar genetic profiles, which warns upon contamination of water sources with clinically significant enterococci. However, diversity in their antibiotic resistance profiles excludes the clonal transmission of bacteria from hospital environment to river water and groundwater. Instead, genetic similarities between freshwater and wastewater strains confirm our hypothesis that anthropic pollution is a major source of antibiotic-resistant enterococci, contributing to their environmental spread. In addition to the enterococcal load, molecular fingerprinting indicates the magnitude of the uncontrolled discharge of untreated or insufficiently treated domestic sewage into the environment. Therefore, ERIC-PCR typing is an improved tool to assess the diversity of *Enterococcus* strains.

This study highlights the importance of water safety in the context of increasing demographic challenge. As a general trend, the population in Cluj is invariably growing, while urbanization and suburbanization affect not only the city infrastructure, but also the surrounding areas. The upstream mountains and isolated hamlets became increasingly popular, as both travel destinations and holiday homes. Recently, especially during the COVID-19 pandemic, another trend has emerged, with counter-urbanization occurring due to changing lifestyles and the opportunity of re-locating work in a home-based office. For the future, an unprecedented enhancement in anthropogenic pressure on water resources is foreseen due to other changes, such as the global warming and the risk of drought. Therefore, the implementation of adequate strategies for the protection of water resources is of paramount importance. Mitigating and adapting to the impacts of demographic change require stringent measures to enforce the regulations for the collection, treatment and discharge of wastewaters in both urban and rural areas. The identification of point sources of pollution, together with the prevention of contamination events are required in order to reduce the microbial risks and to limit the extent of the antibiotic-resistance phenomenon. Proper maintenance of domestic wastewater systems and septic tanks as well as upgrades of municipal sewerage networks and wastewater treatment plants are mandatory. In addition, routine cleaning and disinfection of groundwater wells is effective in the eradication of health hazards associated with the spread of antibiotic-resistant enterococci.

Although this study investigated a large collection of *Enterococcus* isolates and many antibiotic resistance traits, several limitations were identified, including a putative bias in the selection of bacterial isolates and in the investigated ARGs. Therefore, other genetic mechanisms, including novel resistance sequences, could also be responsible for the observed resistance phenotypes. Additional ARGs should be further investigated as more reliable predictors for antimicrobial resistance in environmental enterococci, to eventually elucidate the links between antibiotic resistance and ERIC-PCR genotyping.

## 4. Materials and Methods

### 4.1. Site Description and Sampling Strategy

With a surface of 1603 square kilometers, the Cluj metropolitan area includes Cluj-Napoca city and 19 nearby localities. Due to its dynamics, academic and economic status, and civic and cultural identity, the city constantly attracts new residents. Conducted in 2011, the last official census estimated its population at 324,000 people, while the National Institute of Statistics recorded 740,020 residents living in Cluj County on 1 January 2022 (https://cluj.insse.ro/, accessed on 31 July 2022).

Covering an area of 112 square kilometers, the study site is located in Cluj County, North Western Romania, along the Someșul Mic River basin. The sampling strategy included several types of aquatic environments, sampled in three campaigns: surface waters (SW1, SW2, SW3), groundwater (GW1, GW2, GW3, GW4), municipal wastewaters (WWI and WWE) and hospital effluents (HE). According to their estimated degree of anthropogenic pollution, from low to high, 10 sampling points were set, and a total of 30 water samples were collected. Upstream of the city, two surface waters were sampled near the foothill of Muntele Mare and Munții Gilăului mountains. Tarnița Reservoir (SW1) is a dam reservoir on the Someșul Cald River, the left headwater of Someșul Mic River. With an area of 2.15 square kilometers, a length of more than 8 km and a maximum depth of more than 70 m, it is the main source of drinking water for almost 1 million people. Someșul Rece (SW2), the right headwater of Someșul Mic River, is 49 km long and has a basin size of 330 square kilometers. Downstream of the city, surface water was sampled from Someșul Mic River (SW3), after crossing the city and receiving treated effluents from the municipal wastewater treatment plant. Four sampling points for groundwater were included: an old spring from the peri-urban area (GW1), used for drinking purposes; a shallow well upstream of the city (GW2), near the Someșul Rece River bank; and two hand-dug wells within the city (GW3 and GW4). Wastewater influents (WWI) and effluents (WWE) were sampled from the wastewater treatment plant receiving mainly municipal sewage, as well as hospital input and industrial wastewaters. The plant is designed to process around 115,000 cubic meters of wastewater per day in three-step treatment: mechanical, biological and final deep purification for nitrogen and phosphorus removal. A specialized cancer hospital with 597 beds was selected for the collection of hospital effluents (HE), before sludge treatment and disinfection.

### 4.2. Detection and Enumeration of Intestinal Enterococci

Water samples were collected in sterile recipients and transported in refrigerated boxes into the laboratory. Within 6 h of sampling, microbiological assays were performed for the selective cultivation of intestinal enterococci by direct inoculation or membrane filtration through 0.45 µm sterile membrane filters, according to standard methods (ISO 7899–2:2000. Water quality—Detection and enumeration of intestinal enterococci—Part 2: Membrane filtration method). Red to brown colonies developed on Slanetz Bartley agar (Oxoid, Basingstoke, UK) after 48 h at 37 °C were further confirmed as intestinal enterococci on Bile Esculin Azide Agar (Merck-Millipore, Darmstadt, Germany) by 4 h incubation at 44 °C.

### 4.3. Antimicrobial Susceptibility Testing

Antimicrobial susceptibility testing of 574 isolates was performed using the disk diffusion method described by reference guidelines [54]. Mueller-Hinton agar (Oxoid, Basingstoke, UK) was employed to evaluate bacterial sensitivity to eight antibiotics: ampicillin (2 µg), imipenem (10 µg), norfloxacin (10 µg), gentamicin (30 µg), vancomycin (5 µg), erythromycin (15 µg), tetracycline (30 μg) and trimethoprim-sulfamethoxazole (1.25–23.75 µg). Zone inhibition diameters were interpreted according to clinical breakpoint tables [55,56]. *Enterococcus faecalis* ATCC 2921 was used as a wild-type susceptible strain. Resistance to at least three antimicrobial families was considered multidrug resistance.

### 4.4. ARG Screening

After PCR confirmation with *Enterococcus* molecular markers, 338 bacterial isolates displaying phenotypic resistance were subjected to PCR screening for the detection of ARGs and class 1 integron. Cell suspensions from overnight pure cultures were standardized at 1 MacFarland density. The preparation of DNA templates included freezing, bead beating and boiling procedures for cell wall disruption and enzyme inhibition [57]. DNA amplification was performed in a 15 μL total volume, consisting of 7.5 μL of DreamTaq Green PCR Master Mix (2×) (Thermo Fisher Scientific, Waltham, MA, USA), 0.5 μM of each primer (Eurogentec, Seraing, Belgium), 5.35 μL nuclease-free water (Lonza, Basel, Switzerland), and 2 μL bacterial suspension. PCRs were performed using a TProfessional Trio (Analytik Jena, Jena, Germany) or Mastercycler Nexus (Eppendorf AG, Hamburg, Germany) thermocycler: denaturation 5 min at 94 °C and then 35 cycles of 30 s at 94 °C, 45 s at a specific annealing temperature, 45 s at 72 °C, and a final extension of 8 min at 72 °C. The specific annealing temperatures for each primer pair are given in Table 2. The amplified PCR products were separated in 1.5% *w*/*v* agarose (Cleaver Scientific Ltd., Rugby, UK) gel in 1× TBE buffer (Lonza, Basel, Switzerland) and stained with 0.5 μg/mL ethidium bromide (Thermo Fisher Scientific, Waltham, MA, USA). Data acquisition and interpretation were performed using the BDA Digital Compact System and BioDocAnalyze Software (Analytik Jena, Jena, Germany). Positive and negative controls were included in each set of amplifications. Positive controls used a collection of bacterial strains carrying the targeted genes, previously validated by sequencing.

### 4.5. Molecular Identification

After the phenotypic selection of intestinal enterococci based on standard methods, molecular screening using *Enterococcus* molecular markers [58] was employed to confirm biochemical identification. The bacterial 16S ribosomal RNA gene was used for PCR amplification (Table 2) and subsequent Sanger sequencing for the identification of enterococcal isolates carrying ARGs. Raw sequencing reads were deposited in the GenBank database of National Center for Biotechnology Information (NCBI) under the accession numbers OP359225-OP359304 and OP361300-OP361306. Nucleotide sequences were processed and analyzed using bioinformatic tools available through BioEdit version 7.2, then compared to sequences stored in the GenBank nucleotide database using the blastn algorithm (http://blast.ncbi.nlm.nih.gov/Blast.cgi, accessed on 31 July 2022).

### 4.6. Molecular Fingerprinting of Enterococcus spp.

Repetitive sequence-based polymerase chain reaction (Rep-PCR) was developed using specific ERIC primers involving bacterial DNA suspensions prepared following a protocol previously optimized and demonstrated by Houf et al. [57] as the most efficient for the purpose of ERIC-PCR screening. The ERIC-PCR was carried out with a single primer, which uses the total DNA and, therefore, provides results with good reproducibility [46]. We found that ERIC2 has the greatest discriminatory power among the seven *Enterococcus* species considered in the present study. Reactions were carried out in a total volume of 20 μL containing 10 μL of DreamTaq Green PCR Master Mix (2×) (Thermo Fisher Scientific, Waltham, MA, USA), 1 μM of primer ERIC2 (5′-AAGTAAGTGACTGGGGTGAGCG-3′) (Eurogentec, Seraing, Belgium), 7.8 μL nuclease-free water (Lonza, Basel, Switzerland) and 2 μL template DNA. Amplifications were performed in a TProfessional Trio (Analytik Jena, Jena, Germany) thermocycler with a cycling program consisting of an initial denaturing step at 94 °C for 5 min, 5 cycles of denaturation at 94 °C for 5 min, annealing at 38 °C for 5 min, elongation at 72 °C for 5 min, then 30 cycles of denaturation at 94 °C for 1 min, annealing at 48 °C for 1 min and elongation at 72 °C for 5 min, and a final extension of 72 °C for 10 min. The amplified products were resolved by 1.5% gel electrophoresis at 75 V for 120 min. Data acquisition was performed using the ChemiDoc MP system (BioRad, Hercules, CA, USA) and analyzed with PyElph 1.4 software [59].

### 4.7. Statistical Analysis

Descriptive statistics parameters were applied to assess the mean values and standard deviations of bacterial loads. Proportions, frequencies and patterns of displayed antimicrobial resistance and ARGs were calculated. The relative frequency of ARGs took into account the number of certain gene appearances relative to the total number of ARGs. Statistical correlations between the ERIC-PCR banding patterns and the level of phenotypic and genotypic resistance were inferred using the data analysis tool pack of Microsoft Excel 2016. The heat maps were drawn with CIMminer software, using the quantile-binning method. Quantile divides the weight range of data values into intervals, each with approximately the same number of data points. This effectively spreads out the color differences between data values that are present in regions with a large number of values.

Similarity distances between ERIC-PCR profiles were calculated using the Dice coefficient, and dendrograms were constructed based on the UPGMA analysis with DarWin 6.0.021 software [60]. The PCA multivariate statistical approach was used to explore the effects of ARG variance between different *Enterococcus* species. PCA was executed for the clustering and differentiation of data sets by PAST software version 4.11 [61].

## 5. Conclusions

The outcomes of this study reveal that, besides their role as fecal indicators, intestinal enterococci are hosts for antibiotic resistance determinants that may serve as indicators of anthropogenic impacts on aquatic ecosystems. Rep-PCR fingerprinting using the ERIC2 primer, in conjunction with ARG profiling, is a useful tool for the molecular typing of clinical and environmental *Enterococcus* species. In the context of increasing urbanization and unsustainable human activities in the peri-urban zones, the environmental spread of *Enterococcus* species carrying ARGs is of high concern. Enterococcal release and migration under anthropic pressure leads to the dispersion of clinically relevant strains into the natural environment. These findings support the importance of future strategies for public health protection by defending the water resources. Water quality protection is not only intended to reduce the risk for waterborne outbreaks but also to limit the expansion and progression of the antibiotic resistance phenomenon.

## Figures and Tables

**Figure 1 antibiotics-11-01213-f001:**
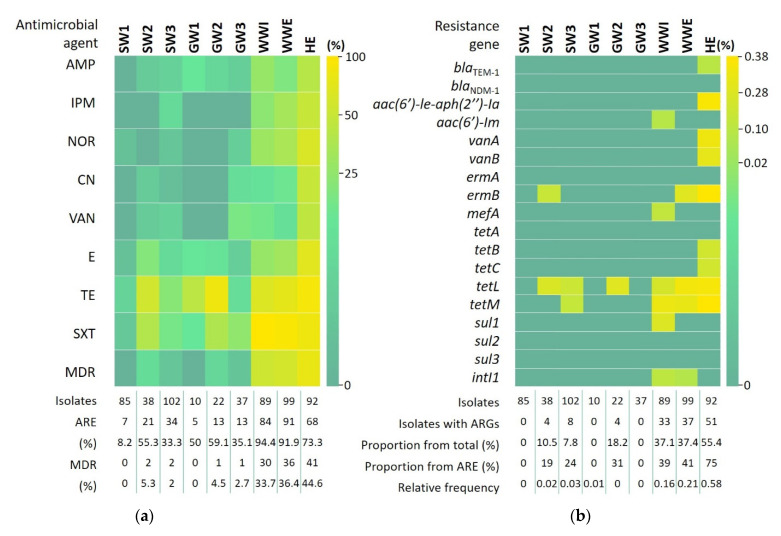
Phenotypic (**a**) and genotypic (**b**) antibiotic resistance of *Enterococcus* spp. isolates. AMP = ampicillin; ARE = antibiotic resistant enterococci; ARGs = antibiotic resistance genes; CN = gentamicin; E = erythromycin; GW = groundwater; HE = hospital effluent; IPM = imipenem; MDR = multidrug resistance; NOR = norfloxacin; SXT = trimethoprim-sulfamethoxazole; SW = surface water; TE = tetracycline; VAN = vancomycin; WWE = wastewater effluent; WWI = wastewater influent. Quantile binning method was applied for both heat maps construction. The color code uses green for low values to yellow for high values.

**Figure 2 antibiotics-11-01213-f002:**
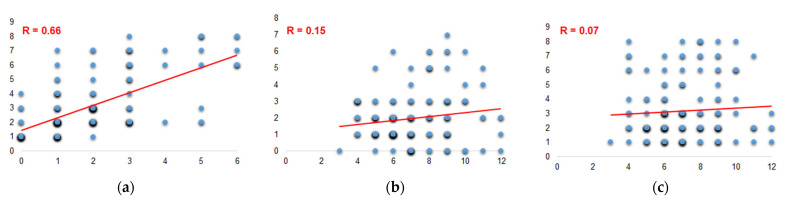
Scatter plots showing linear regression lines and correlation coefficients between the levels of phenotypic and genotypic resistance (**a**), the number of ERIC-PCR bands and phenotypic resistance (**b**), the number of ERIC-PCR bands and ARGs detected in *Enterococcus* isolates (**c**).

**Figure 3 antibiotics-11-01213-f003:**
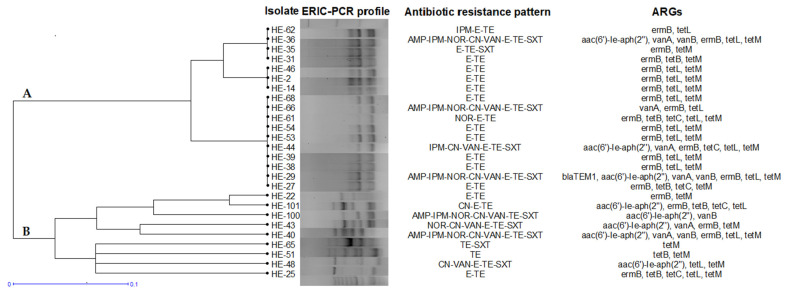
ERIC-PCR dendrogram and antibiotic resistance profiles of *E. avium* isolates. All isolates were from the hospital effluent.

**Figure 4 antibiotics-11-01213-f004:**
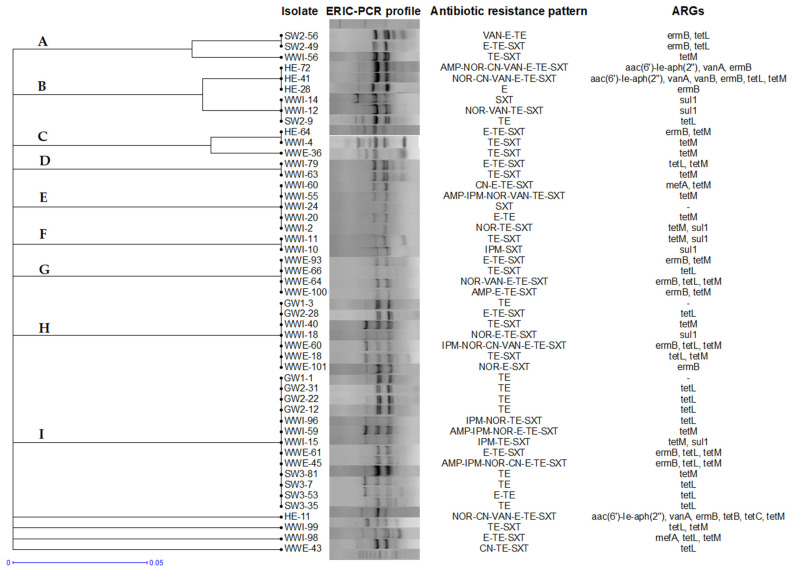
ERIC-PCR dendrogram and antibiotic resistance profiles of *E. faecalis* isolates. The isolates were labelled by sources: GW = groundwater; HE = hospital effluent; SW = surface water; WWI = wastewater influent; WWE = wastewater effluent.

**Figure 5 antibiotics-11-01213-f005:**
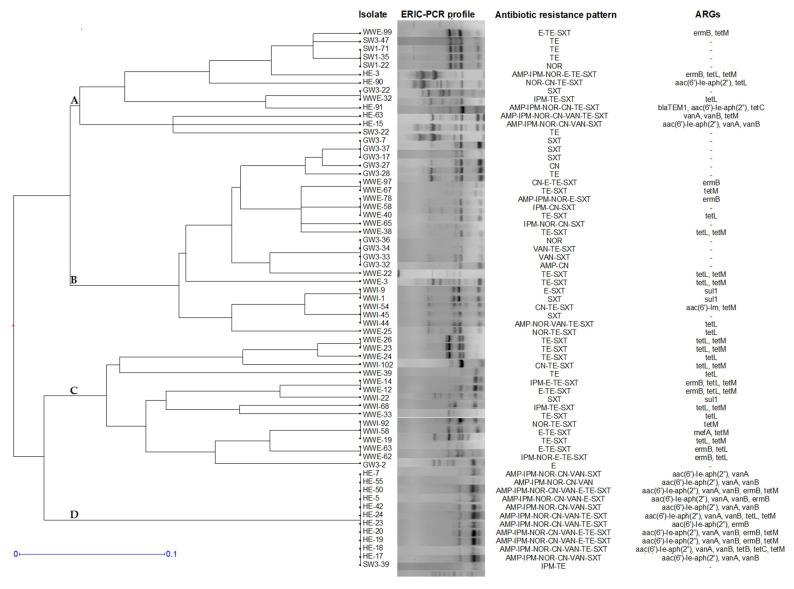
ERIC-PCR dendrogram and antibiotic resistance profiles of *E. faecium* isolates. The isolates were labelled by sources: GW = groundwater; HE = hospital effluent; SW = surface water; WWI = wastewater influent; WWE = wastewater effluent.

**Figure 6 antibiotics-11-01213-f006:**
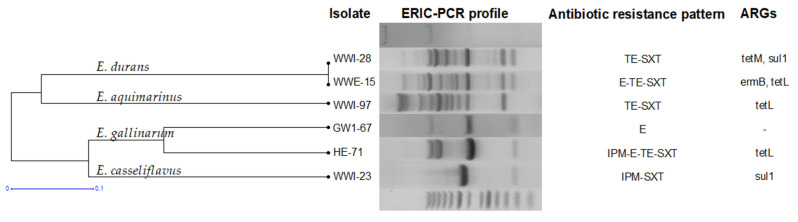
ERIC-PCR dendrogram and antibiotic resistance profiles of other *Enterococcus* spp. The isolates were labelled by sources: GW = groundwater; HE = hospital effluent; WWI = wastewater influent; WWE = wastewater effluent.

**Figure 7 antibiotics-11-01213-f007:**
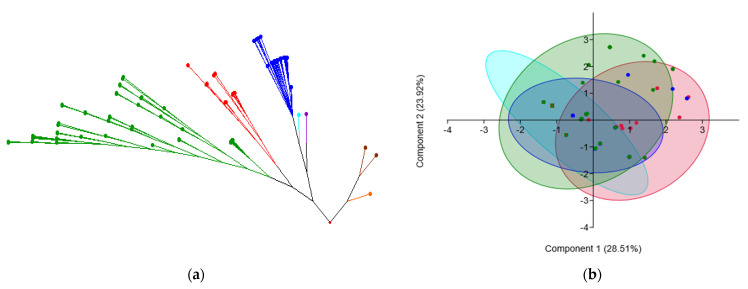
Molecular typing revealing clustering of *Enterococcus* spp. by: (**a**) ERIC-PCR profiles; (**b**) ARG patterns. Color codes: *E avium* (red); *E. faecalis* (blue); *E. faecium* (green); *E. aquimarinus* (aqua); *E. casseliflavus* (orange); *E. durans* (purple); *E. gallinarum* (brown). In (**b**) PCA clustering of non-predominant species appears in aqua.

**Table 1 antibiotics-11-01213-t001:** Contamination of water by enterococci along the aquatic compartments.

Parameter	SW1	SW2	SW3	GW1	GW2	GW3	GW4	HE	WWI	WWE
Intestinal enterococci (CFU/100 mL)	9 ± 1	43 ± 5	(11.7 ± 0.1) × 10^3^	3 ± 2	80 ± 6	12 ± 1	0	(18 ± 0.1) × 10^3^	(465 ± 0.2) × 10^3^	99 ± 5
No. of tested isolates	85	38	102	10	22	37	0	92	89	99
No. of identified isolates	3	3	7	3	4	11	0	48	33	34
*E. aquimarinus*	0	0	0	0	0	0	0	0	1	0
*E. avium*	0	0	0	0	0	0	0	26	0	0
*E. casseliflavus*	0	0	0	0	0	0	0	0	1	0
*E. durans*	0	0	0	0	0	0	0	0	1	1
*E. faecalis*	0	3	4	2	4	0	0	5	20	11
*E. faecium*	3	0	3	0	0	11	0	16	10	22
*E. gallinarum*	0	0	0	1	0	0	0	1	0	0

Note: CFU = colony forming units; GW = groundwater; HE = hospital effluent; SW = surface water; WWE = wastewater effluent; WWI = wastewater influent.

**Table 2 antibiotics-11-01213-t002:** Primers used for PCR amplifications.

No.	Target Gene	Primer Sequence (5′–3′)	Amplicon (bp)	Annealing Temperature	NCBI Reference Sequence
1	*bla* _TEM-1_	GGTCGCCGCATACACTATTC/ ATACGGGAGGGCTTACCATC	500	57 °C	AL513383.1
2	*bla* _NDM-1_	GGTTTGGCGATCTGGTTTTC/ CGGAATGGCTCATCACGATC	52	55 °C	HQ256747.1
3	*aac(6′)-Im*	GGCTGACAGATGACCGTGTTCTTG/ GTAGATATTGGCATACTACTCTGC	303	53 °C	NG_052530.1
4	*aac(6′)-Ie-aph(2”)-Ia*	CCAAGAGCAATAAGGGCATA/ CACTATCATAACCACTACCG	220	51 °C	KM083808.1
5	*vanA*	GCTATTCAGCTGTACT/ CAGCGGCCATCATACGG	783	51 °C	M97297.1
6	*vanB*	CGCCATACTCTCCCCGGATAG/ AAGCCCTCTGCATCCAAGCAC	667	61 °C	KF823969.1
7	*ermA*	GAACCAGAAAAACCCTAAAGACAC/ ACAGAGTCTACACTTGGCTTAGGATG	507	57 °C	X03216.1
8	*ermB*	GAAAAGGTACTCAACCAAAT/ AGTAACGGTACTTAAATTGTTTAC	639	50 °C	AY827541.1
9	*mefA*	CATCGACGTATTGGGTGCTG/ CCGAAAGCCCCATTATTGCA	455	55 °C	AY071835.1
10	*tetA*	GCAAGCAGGACCATGATCGG/ GCCGATATCACTGATGGCGA	572	57 °C	AF534183.1
11	*tetB*	GGTTAGGGGCAAGTTTTGGG/ ATCCCACCACCAGCCAATAA	541	57 °C	NG_048168.1
12	*tetC*	TGAGATCTCGGGAAAAGCGT/ AAAGCCGCGGTAAATAGCAA	460	53 °C	NC_024960.1
13	*tetL*	TATTCAAGGGGCTGGTGCAG/ CGGCAGTACTTAGCTGGTGA	545	57 °C	AY081910.1
14	*tetM*	CCGTCTGAACTTTGCGGAAA/ CAACGGAAGCGGTGATACAG	627	57 °C	AJ585076.1
15	*sul1*	AGGCATGATCTAACCCTCGG/ GGCCGATGAGATCAGACGTA	665	57 °C	JF969163.1
16	*sul2*	GACAGTTATCAACCCGCGAC/ GAAACAGACAGAAGCACCGG	380	57 °C	AY055428.1
17	*sul3*	GTGGGCGTTGTGGAAGAAAT/ AAAAGAAGCCCATACCCGGA	370	57 °C	FJ196385.1
18	*intI1*	CCTGCACGGTTCGAATG/ TCGTTTGTTCGCCCAGC	497	55 °C	NZ_JAMYXD010000016.1
19	*16S rRNA*	AGAGTTTGATCCTGGCTCAG/ ACGGCTACCTTGTTACGACTT	1519	56 °C	AB012212.1
20	*16S Enterococcus*	GGACGMAAGTCTGACCGA/ TTAAGAAACCGCCTGCGC	221	57 °C	JQ804949.1

## Data Availability

Raw sequencing reads were deposited in the GenBank database of National Center for Biotechnology Information (NCBI) under the accession numbers OP359225-OP359304 and OP361300-OP361306.

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
