# Peer review of "Molecular Typing Reveals Environmental Dispersion of Antibiotic-Resistant Enterococci under Anthropogenic Pressure"

_antibiotics, 2022, doi:10.3390/antibiotics11091213_

Round 1

Reviewer 1 Report

The manuscript entitled "Molecular typing reveals environmental dispersion of antibiotic-resistant enterococci under anthropogenic pressure" highlights the contamination of fecal microbes as a result of anthropogenic pressure. Authors have revealed phenotypic resistance along with antimicrobial resistance determinants or ARGs in Enterococci along with their genetic diversity in different water sources.

Overall, the article is well written and provides significantly important information and data to the scientific community around the globe. However, I have a few comments to be addressed. 

1. In the methodology, the authors didn't provide clear information regarding the total number of samples collected. 

2. I presumed that during ARG screening, the authors didn't perform DNA extraction and purification. What influence do they cause on PCR performance as in other cases there are certain PCR inhibitory substances in bacterial cells that can intervene in PCR reaction? 

3. I suggest adding a few sentences in discussion or in conclusion regarding the public health impacts of the results found in this study. 

Author Response

Thank you very much for your support. We are extremely grateful  for the attention you have given to our study and requesting for a revised version. The manuscript was improved by resolving all comments as responded below:

Comment: In the methodology, the authors didn't provide clear information regarding the total number of samples collected.

Response: A more detailed description was added in the sampling subsection.

Comment: I presumed that during ARG screening, the authors didn't perform DNA extraction and purification. What influence do they cause on PCR performance as in other cases there are certain PCR inhibitory substances in bacterial cells that can intervene in PCR reaction? 

Response: We agree reviewer’s suggestion; we did not explained sufficiently the steps for lysed bacteria processing. The protocol for DNA template preparation and control methods were detailed in the ARG screening subsection. As a comment, we would like to add that during the past years, and even within the EnviroAMR project, our research team applied multiple variations to different protocols for optimizing the sensitivity and accuracy of rapid detection of ARGs by PCR in bacteria from environmental samples. A methodological guide was issued for monitoring antibiotic residues and antimicrobial resistance in the environment, as a supporting instrument for an enhanced quality management of surface waters and groundwater.

Comment: I suggest adding a few sentences in discussion or in conclusion regarding the public health impacts of the results found in this study.

Response: Discussion, conclusions and abstract sections were improved by resuming the public health implications and adding suggestions for the practical implementation of our findings.

Reviewer 2 Report

dear authors, can you take in your considerations the following criticisms for the manuscript entitled" 

Molecular typing reveals environmental dispersion of antibiotic resistant enterococci under anthropogenic pressure"

Title: the title must be more informative and attractive

 Keywords: Key words of the manuscript are too long and uninformative, you must concise them.

 Abstract section

· Line 26 you cannot start a new sentence with abbreviation “E. faecium” and also in line 28

·The sentence in line 26 is incorrect please rephrase this sentence

·       Please add the overall practical implementation of your results or other hypothesis that may be utilized in the future at the end of the abstract. Such as the over aim for the environmental dispersion of antibiotic resistant enterococci.

Introduction section

·       The introduction section was too long and focused on the spreading of enterococci and doesn’t discuss the resistance genes and the different ways for their spreading

·       Throughout the manuscript, many punctuations were incorrect, please carefully revise, for example “line 51”

·       Please if you can to elaborate  in detail the worldwide antibiotic resistant crises

·       The aim of this study must be rephrased you cannot used the tittle of this repost  as the sole  aim

Results

·       Throughout the manuscript and in the tittle you have to replace the word “antibiotic” by “antimicrobial agent” due to the using of SXT

·       Please provide the accession number for the sequenced genes 

·       Please provide the color code description in the figure ligand 1

·       Can you divid the figure 2,3,4 into 2 part, first (A):  ERIC-PCR only and another part (B); which showed the diversity of these isolates based on both ERIC-PCR, phenotypic and genotypic antimicrobial resistance then describe the discriminatory power of these combined typing methods

·       Can you add figure for significant correlation which showed the result in line 193

·       Where is the result of ermA gene

·       The scientific written of the abbreviation of the tested genes throughout the manuscript need to be revised “When referring to the genetic element (or genotype), the abbreviation must be italicized and the first 3 letters are lowercase” as in line 360

Discussion:

·       It needs numerous modifications. It should focus on explaining and evaluating what you found (the main results), showing how it relates to the new researches

 Methodology:

·       Can you provide the references for the primer and cycle conditions for all tested antimicrobial agents and other genes

Conclusion section:

·       It must be rephrased, conclusion section must provide us with the applied implication of your results in concise manner

·       Please add at the end of the manuscript the limitations: what can’t the results and discussion tell us?

Author Response

Thank you for considering our manuscript and requesting for a revised version.  We hope we found the inspiration to resolve all comments and enhance the quality and the presentation of the manuscript as suggested. All comments were addressed and the responses are individually listed below:

Comment: the title must be more informative and attractive

R: The tile was slightly adjusted.

Comment: Key words of the manuscript are too long and uninformative, you must concise them.

R: Four keywords were reconsidered and adapted. If the reviewer is referring also to the enumeration of the three species (Enterococcus avium; Enterococcus faecalis; Enterococcus faecium) we decided to keep it as listed, justified by its relevance. Previous investigations were often focused on E. faecium or E. faecalis and sometimes on both species. To our knowledge, there are no studies published so far to fingerprint E. avium, at least by ERIC-PCR.

Comments: Line 26 you cannot start a new sentence with abbreviation “E. faecium” and also in line 28. The sentence in line 26 is incorrect please rephrase this sentence

R: We agree, the suggestion was implemented and the sentence was corrected.

Comment: Please add the overall practical implementation of your results or other hypothesis that may be utilized in the future at the end of the abstract. Such as the over aim for the environmental dispersion of antibiotic resistant enterococci.

R: The abstract was improved by resuming the public health implications and adding suggestions for the practical implementation of our findings.

Introduction

Comment: The introduction section was too long and focused on the spreading of enterococci and doesn’t discuss the resistance genes and the different ways for their spreading

R: We agree, this manuscript was designed to evaluate the environmental spreading of Enterococcus strains. Therefore, in the introduction, a state-of the art for antibiotic resistance was presented in relation to Enterococus, and the relevance of different ARGs was further detailed in the discussion section.

Comment: Throughout the manuscript, many punctuations were incorrect, please carefully revise, for example “line 51”

R: We have reviewed the manuscript and carefully modified the punctuations.

Comment: Please if you can to elaborate in detail the worldwide antibiotic resistant crises

R: We agree the worldwide antibiotic crisis is an urgent global public health, and our manuscript falls within this general topic. Specifically, the role of aquatic environments as matrices for antimicrobial resistant bacteria and their ARGs was presented in the second paragraph of the introduction section. Moreover, the particular role of enterococci as leading causes of multidrug-resistant hospital-acquired infections threatening the therapeutic effects of antibiotics in use was described in the fourth paragraph.

Comment: The aim of this study must be rephrased you cannot used the tittle of this repost  as the sole  aim

R: As the title of the manuscript was rephrased, we would suggest keeping the elaborate aim of the study as it was designed. Instead, specific objectives were added for a better understanding of our purpose.

Results

Comment: Throughout the manuscript and in the tittle you have to replace the word “antibiotic” by “antimicrobial agent” due to the using of SXT

R: We appreciate this suggestion, the terminology was corrected through the manuscript.

Comment: Please provide the accession number for the sequenced genes 

R: We recognize the importance of data availability. The sequences for 16S rDNA were submitted to the GenBank archive of NCBI under the submission number SUB11966687 and still awaiting for the publication.

Comment: Please provide the color code description in the figure ligand 1

R: The suggestion was implemented and the color scale was explained.

Comment: Can you divid the figure 2,3,4 into 2 part, first (A):  ERIC-PCR only and another part (B); which showed the diversity of these isolates based on both ERIC-PCR, phenotypic and genotypic antimicrobial resistance then describe the discriminatory power of these combined typing methods

R: We recognize that these figures are large. Unfortunately they undergone few tentative changes as suggested, but splitting them would make them even larger but not more explanatory. We decided to keep them as they are, because we found this format very informative due to its complexity. This type of representation is often used to display relationships between strains with different sources of origin, locations, and phenotypic, biochemical, metabolic or genetic traits. The comparative and combined discriminatory power of these methods was discussed in text.

Comment: Can you add figure for significant correlation which showed the result in line 193

R: Scatter plots and correlation coefficients were displayed in a new figure.

Comment: Where is the result of ermA gene

R: As displayed in Figure 1b and mentioned in text, PCR amplifications for blaNDM-1, ermA, tetA, sul2 and sul3 had negative results.

Comment: The scientific written of the abbreviation of the tested genes throughout the manuscript need to be revised “When referring to the genetic element (or genotype), the abbreviation must be italicized and the first 3 letters are lowercase” as in line 360

R: We have changed the formatting of ARGs, except for blaTEM-1 and blaNDM-1, which were abbreviated according to the Consensus on β-Lactamase Nomenclature (Bradford et al. Antimicrob Chemother 2022).

Discussion:

Comment: It needs numerous modifications. It should focus on explaining and evaluating what you found (the main results), showing how it relates to the new researches

R: Discussion section was improved and reorganized to summarize the important findings, highlight the interesting results and integrate them in the context of current research. Also, the public health implications regarding the environmental spread of resistant enterococci and suggestions for the practical implementation of our findings were brought in discussion.

Paragraph 1: Summary of the project

Paragraph 2: Evaluates the role of intestinal enterococci as faecal indicators discussing the prevalence of enterococcus species in different aquatic matrices related to anthropogenic pressure and compares the results to current research in the field.

Paragraph 3: Compares results of antibiotic resistance traits and their significance in different aquatic matrices

Paragraph 4: Describes the mechanisms of encoded resistance, compares antibiotic resistance traits found in different Enterococcus species and discusses their role in the spread of resistance genes

Paragraph 5: Discusses the ERIC-PCR typing method, its discriminatory power between Enterococcus species and strains and highlights its relevance for assessing genetic similarities in isolates from different sources as an improved tool to investigate diversity of Enterococcus strains.

Paragraph 6: Highlights the importance of water safety strategies in the context of increasing anthropogenic pressure

Paragraph 7: Details some of the limitations of the study

Methodology:

Comment: Can you provide the references for the primer and cycle conditions for all tested antimicrobial agents and other genes

R: During methods optimization, all the primers used in this study were verified and validated against reference sequences as specified in Table 2. CARD database was used for ARGs sequences, then ARG sequences from Enterococcus spp. were selected from NCBI database, where available. References for Enterococcus molecular markers and ERIC primers were provided in text. As mentioned, every PCR program was designed individually and optimized during this study.

Conclusion section:

Comment: It must be rephrased, conclusion section must provide us with the applied implication of your results in concise manner

R: We agree, conclusion section was amended with emphasis on practical implications of the study.

Comment: Please add at the end of the manuscript the limitations: what can’t the results and discussion tell us?

R: Limitations were identified and included in the end of discussion.

Round 2

Reviewer 2 Report

comments to authors 

Old Comment: Please provide the accession number for the sequenced genes

Response: We recognize the importance of data availability. The sequences for 16S rDNA were submitted to the GenBank archive of NCBI under the submission number SUB11966687 and still awaiting for the publication.

New comment; Please can you add the accession number for 16S rRNA gene which was provided during the submission to gene bank  after the submission by 3 or 4 days ? I know that this accession number take long time to be release by gene bank; however the accession number must be added in the manuscript

Old Comment: The introduction section was too long and focused on the spreading of enterococci and doesn’t discuss the resistance genes and the different ways for their spreading

Response: We agree, this manuscript was designed to evaluate the environmental spreading of Enterococcus strains. Therefore, in the introduction, a state-of the art for antibiotic resistance was presented in relation to Enterococus, and the relevance of different ARGs was further detailed in the discussion section.

New comment; The worldwide antibiotic resistance crises must be well discussed in detail in the introduction section and discussion

Old Comment: the title must be more informative and attractive;

Response: The tile was slightly adjusted.

New comment: however the author replace the word antibiotic by antimicrobial only

Old Comment: Can you divid the figure 2,3,4 into 2 part, first (A): ERIC-PCR only and another part (B); which showed the diversity of these isolates based on both ERIC-PCR, phenotypic and genotypic antimicrobial resistance then describe the discriminatory power of these combined typing methods

Response: We recognize that these figures are large. Unfortunately they undergone few tentative changes as suggested, but splitting them would make them even larger but not more explanatory. We decided to keep them as they are, because we found this format very informative due to its complexity. This type of representation is often used to display relationships between strains with different sources of origin, locations, and phenotypic, biochemical, metabolic or genetic traits. The comparative and combined discriminatory power of these methods was discussed in text.

New comments: still recommended to show the discriminatory power for ERIC-PCR alone therefore these figure should be divided in 2 parts  

Author Response

Thank you for your prompt response. We have decided to keep the manuscript in its present form. Please consider the rationale for not implementing additional changes:

  1. Old Comment: Please provide the accession number for the sequenced genes

Response: We recognize the importance of data availability. The sequences for 16S rDNA were submitted to the GenBank archive of NCBI under the submission number SUB11966687 and still awaiting for the publication.

New comment; Please can you add the accession number for 16S rRNA gene which was provided during the submission to gene bank  after the submission by 3 or 4 days ? I know that this accession number take long time to be release by gene bank; however the accession number must be added in the manuscript

New response: Accession number for the sequenced genes are under publication in NCBI Genbank. Once published, they will be linked to the article, but accession numbers are not available at this moment, nor will be in 3-4 days.

  1. Old Comment: The introduction section was too long and focused on the spreading of enterococci and doesn’t discuss the resistance genes and the different ways for their spreading

Response: We agree, this manuscript was designed to evaluate the environmental spreading of Enterococcus strains. Therefore, in the introduction, a state-of the art for antibiotic resistance was presented in relation to Enterococus, and the relevance of different ARGs was further detailed in the discussion section.

New comment; The worldwide antibiotic resistance crises must be well discussed in detail in the introduction section and discussion

New response: Discussion about the antibiotic resistance crisis in the introduction. We do not find appropriate to add more details on this general topic. The manuscript already has almost 10.000 words.

  1. Old Comment: the title must be more informative and attractive;

Response: The tile was slightly adjusted.

New comment: however the author replace the word antibiotic by antimicrobial only

New response: The title must be more informative and attractive. We do not fully understand this request. After few tentative changes we haven’t found a better alternative and decided to keep the manuscript title as is.

  1. Old Comment: Can you divid the figure 2,3,4 into 2 part, first (A): ERIC-PCR only and another part (B); which showed the diversity of these isolates based on both ERIC-PCR, phenotypic and genotypic antimicrobial resistance then describe the discriminatory power of these combined typing methods

Response: We recognize that these figures are large. Unfortunately they undergone few tentative changes as suggested, but splitting them would make them even larger but not more explanatory. We decided to keep them as they are, because we found this format very informative due to its complexity. This type of representation is often used to display relationships between strains with different sources of origin, locations, and phenotypic, biochemical, metabolic or genetic traits. The comparative and combined discriminatory power of these methods was discussed in text.

New comments: still recommended to show the discriminatory power for ERIC-PCR alone therefore these figure should be divided in 2 parts.

New response: Figures splitting. As we explained, we do not agree to split 4 figures in 8 figures and loose the informative content. Or maybe we don’t fully understand this comment. I must underline that this format is often used in articles comparing the dedrograms of typing results with antibiotic resistance or virulence traits